# $\alpha$-Clustering in atomic nuclei from first principles with statistical learning and the Hoyle state character

T. Otsuka [1,2,3 ✉], T. Abe [2,4], T. Yoshida[4,5], Y. Tsunoda [4], N. Shimizu[4], N. Itagaki[6], Y. Utsuno [3,4], J. Vary [7], P. Maris [7] & H. Ueno[2]

A long-standing crucial question with atomic nuclei is whether or not $\alpha$ clustering occurs there. An $\alpha$ particle (helium-4 nucleus) comprises two protons and two neutrons, and may be the building block of some nuclei. This is a very beautiful and fascinating idea, and is indeed plausible because the $\alpha$ particle is particularly stable with a large binding energy. However, direct experimental evidence has never been provided. Here, we show whether and how $\alpha$(-like) objects emerge in atomic nuclei, by means of state-of-the-art quantum many-body simulations formulated from first principles, utilizing supercomputers including K/Fugaku. The obtained physical quantities exhibit agreement with experimental data. The appearance and variation of the $\alpha$ clustering are shown by utilizing density profiles for the nuclei beryllium-8, -10 and carbon-12. With additional insight by statistical learning, an unexpected crossover picture is presented for the Hoyle state, a critical gateway to the birth of life.

[1] Department of Physics, The University of Tokyo, 7-3-1 Hongo, Bunkyo, Tokyo 113-0033, Japan. [2] RIKEN Nishina Center, 2-1 Hirosawa, Wako, Saitama 351-0198, Japan. [3] Advanced Science Research Center, Japan Atomic Energy Agency, Tokai, Ibaraki 319-1195, Japan. [4] Center for Nuclear Study, The University of Tokyo, 7-3-1 Hongo, Bunkyo, Tokyo 113-0033, Japan. [5] Research Organization for Information Science and Technology, 2-4, Shirakata, Tokai, Ibaraki 319-1106, Japan. [6] Yukawa Institute for Theoretical Physics, Kyoto University, Kitashirakawa Oiwake-Cho, Kyoto 606-8502, Japan. [7] Department of Physics and Astronomy, Iowa State University, Ames, IA 50011, USA. ✉email: otsuka@phys.s.u-tokyo.ac.jp

The atomic nucleus comprises $Z$ protons and $N$ neutrons, which are collectively called nucleons. In the $\alpha$ clustering picture as illustrated in Fig. 1, the $\alpha$ particle ($Z = N = 2$) forms a building block, and some nuclei can be composed of $\alpha$ particles. In such cases, $Z = N = 2i$ holds with $i$ being an integer, and the mass number $A = Z + N$ becomes equal to 4, 8, 12, ... A given nucleus is labeled as $^{A}X$ where X denotes the element, e.g., $^{8}$Be for beryllium-8. Fig. 1b–c sketch intuitive pictures for possible $\alpha$ clustering in $^{8}$Be and $^{12}$C, respectively, where $\alpha$ particles are shown by mid-sized circles forming nuclei represented by green areas. Such natural pictures, collectively called the $\alpha$ cluster model, have been conceived since the 1930s[1–7]. It is, however, still difficult to observe the $\alpha$ clustering experimentally. This is basically because the nucleus is not at rest (quantum mechanically) but we need its snapshot (see Fig. 1).

An alternative possibility is theoretical studies: quite a few studies, for example[8–16], were performed based on models or assumptions including limiting cases like linear chains[3,15], equilateral triangles[13], and a Bose-Einstein condensate[14]. More recently ab initio calculations were reported[17–20], where two $\alpha$ clusters in the ground state of $^{8}$Be were suggested[17] (see Fig. 1b). The $\alpha$ clustering is more crucial but less clarified for the $^{12}$C nucleus: this nucleus can be formed by three $\alpha$ particles in configurations, triangular, linear, or other (see Fig. 1c). Its lowest spin/parity $J^{\pi} = 0^{+}$ excited state, the infamous Hoyle state[21–23], is a critical gateway in the nucleosynthesis to the present carbon-abundant world filled with living organisms[24,25], but its structure remains to be clarified.

We show in this work, by state-of-the-art computational simulations without assuming $\alpha$ clustering a priori, that $\alpha$ clustering indeed occurs for the ground and excited states of $^{8,10}$Be and $^{12}$C isotopes, including the Hoyle state, in varying formation patterns. The simulations are performed by full Configuration-Interaction (CI) calculations from first principles on a sound basis, and their validity is further examined for some observables by comparing with experimental data. The revealed features are supported by a statistical learning technique[26], and present an unexpected crossover[27] between clustering and normal nuclear matter.

## Results

**Multi-nucleon structure by CI simulation.** The present CI calculation is called the shell-model (SM) calculation in nuclear physics. Among various types of SM calculations, the one taken in this work belongs to Monte Carlo Shell Model (MCSM)[28–31]. The MCSM has already been applied to various studies on atomic nuclei (see examples, [32,33]). The present MCSM calculation differs in that all nucleons are activated (i.e., no inert core)[34,35], implying no core-polarization (or in-medium) correction is needed. The nucleon-nucleon ($NN$) interaction is fixed on a fundamental basis prior to this work as described below, so as to accurately describe free $NN$ scattering[36–38]. The whole scheme can then be referred to as the ab initio No-Core MCSM, which is

a state-of-the-art CI calculation for nuclei running on supercomputers such as K[39] and Fugaku[40].

The $NN$ interaction we use is the JISP16 interaction[36] for Be isotopes and the Daejeon16 interaction[37] for C isotopes. The inter-nucleon potential of the JISP16 interaction was determined so as to reproduce $NN$ scattering data and deuteron properties. In addition, the binding energies of light nuclei are used for fine-tuning. No explicit three-nucleon interactions are included, but momentum-dependent $NN$ interaction terms produce similar effects[36]. The Daejeon16 interaction is a successor of JISP16. It has been derived from chiral effective field theory up to N3LO terms[38], and also uses a few properties of light nuclei for the fine-tuning instead of three-nucleon forces[37]. Both interactions have been fixed prior to the present simulation and retain their excellent descriptions of the $NN$ scattering data. For the Be isotopes, the results of JISP16 interaction are used in this paper, because of no notable change by Daejeon16.

In the present CI calculations, protons and neutrons are moving in certain single-particle states, taking various configurations. Their many-body structure is obtained as solutions of the Schrödinger equation with the aforementioned $NN$ interaction. These single-particle states are given by eigenstates of the harmonic-oscillator (HO) potential. We take a sufficiently large number of such eigenstates so that a good accuracy is achieved: the HO shells up to the 6th ($5\hbar\omega$) or 7th shell ($6\hbar\omega$) for Be and C isotopes, respectively, with $\hbar\omega$ being the HO quantum. We note that the present simulation employs cutting-edge supercomputing: if we were to attempt the same calculation with direct matrix diagonalization, the dimension of the vector space is as large as $1.2 \times 10^{12}$ for $^{8}$Be and $1.9 \times 10^{19}$ for $^{12}$C. The MCSM enables us to solve the Schrödinger equation to a good approximation[34], without resorting to such formidable calculations. Some of the ground-state properties obtained by the present calculation are reported elsewhere[35], and we shall here focus on the clustering.

**Manifestation of $\alpha$-clustering and beryllium isotopes.** The aforementioned eigensolutions provide energy eigenvalues and wave functions. Figure 2 indicates, for $^{8,10,12}$Be, the excitation energies, $E_x(J^{\pi})$, of the states of $J^{\pi} = 2^{+}$ or $4^{+}$ on top of the $J^{\pi} = 0^{+}$ ground state, while other excited states are omitted for clarity. One sees a good agreement between the present CI simulation and experiment. Because this simulation is a first-principles calculation with no adjustable parameters, this agreement deserves particular attention. Similar results were obtained for $^{8}$Be by the Green's Function Monte Carlo (GFMC) calculation[17,18], and for Be isotopes by the no-core CI calculation with the JISP16 interaction[41]. The three isotopes in Fig. 2 commonly exhibit a pattern $E_x(4^{+})/E_x(2^{+}) \sim 3$, as reproduced rather well by the present work. This is a typical pattern of the rotational motion of a non-spherical quantum object. A schematic image of the rotational motion of a di-cluster formation is displayed in Fig. 1b.

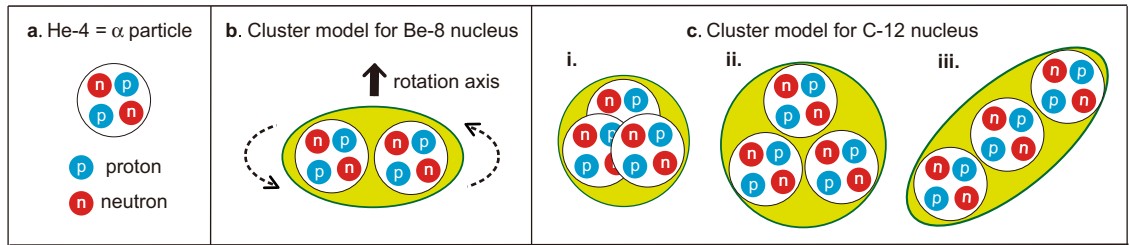

**Fig. 1 Schematic illustrations of $\alpha$ clustering in atomic nuclei. a** $^{4}$He$=\alpha$ particle, **b** $^{8}$Be, and **c** $^{12}$C (three possible cases, i, ii, and iii). The green areas represent atomic nuclei allowing some movements of $\alpha$ clusters.

As the motion of all nucleons is explicitly treated, we calculate the density distributions of protons and neutrons from the wave functions. The density of protons and neutrons combined is called matter density. The ground state of the $^8$Be nucleus has $J^\pi = 0^+$, and hence the matter density should be isotropic in the laboratory frame. Its theoretical density is shown in Fig. 3c. From the measurement of this density, even if this were feasible, it is virtually impossible to extract a footprint of the $\alpha$ clustering. In fact, what we need is an instantaneous "snapshot" of the density distribution (see Fig. 1b), but the experiment cannot provide it yet. However, we show here how to develop a theoretical snapshot from the MCSM wave function. It has been known that the rotational band members, like those stated above, can be described by a single snapshot, which is a particular state in the body-fixed frame and rotates with given spins $J = 0, 2, \ldots$ in the laboratory frame[42]. We extract, in this work, this snapshot state from the MCSM calculation. (Note that although the snapshot state is often called the intrinsic state, we prefer snapshot state for the sake of clarity).

The wave functions of the MCSM are expanded with so-called MCSM basis vectors, which are deformed Slater determinants. Some 50–200 MCSM basis vectors are selected according to their contributions to the energy eigenvalue, from a much larger number of candidates generated stochastically. Such selected basis vectors are further refined by variational recipes. One of the useful features of MCSM is that each basis vector carries a certain

character expressed by its density profile, as exemplified in Fig. 3g–j for the $^8$Be ground state. These density profiles are originally in random directions, but their orientations are now aligned. In this alignment process, we first calculate and diagonalize the matter quadrupole matrix of each basis vector, yielding three eigenvalues of the quadrupole moment. Such quadrupole moments, obtained within quantum mechanics, are mapped onto a classical uniform-density ellipsoid with the same quadrupole moments. As shown in Fig. 4a, this ellipsoid is specified by the three (principal) axes: the longest, middle, and shortest axes, $R_z$, $R_y$, and $R_x$, put on the $z$, $y$, and $x$ coordinates.

We then introduce the *Q-aligned state*: all basis vectors are aligned so that $R_z$, $R_y$, and $R_x$ of each basis vector point to the pre-fixed directions. After this alignment process, all basis vectors are superposed with their calculated amplitudes so as to yield the MCSM ground state when the Q-aligned state is projected onto $J^\pi = 0^+$ (or equivalently the $J^\pi = 0^+$ component is extracted). The snapshot state we seek from theory is given by this Q-aligned state, which provides the "snapshot of density profile". We note that the Q-aligned state indeed describes the rotational bands almost perfectly, as confirmed by the angular momentum projection performed numerically. This is remarkable since the outcome from the first principles exhibits the rotational feature, which is classical in the sense that a fixed object rotates. (For technical details, see Methods.)

We now move on to actual density profiles of $^8$Be. Figure 3d displays the density profile of the Q-aligned state of $^8$Be on the $xz$ plane, visualizing two $\alpha$ clusters. For comparison, Fig. 3b depicts the density profile of the $\alpha$ particle, which agrees with experiment[43] and is spherical (implying no difference between laboratory and body-fixed frames). The $^8$Be density on the $xy$ plane is shown in panel e (f) for $z = 1.65$ (0.0) fm; the point of highest density is near the former plane. These panels display almost circular patterns with different magnitudes, consistent with the $\alpha$ clustering. The $\alpha$ clustering thus emerges out of the first-principles calculation without any assumption nor built-in constraint. It is remarkable that in Fig. 3d, the two clusters look like two free $\alpha$ particles with a separation distance ~3.5 fm, while slight prolongation along the $z$ axis is seen. The Variational Monte Carlo calculation produced a similar density distribution in a different manner[17]. Although the present calculation is made

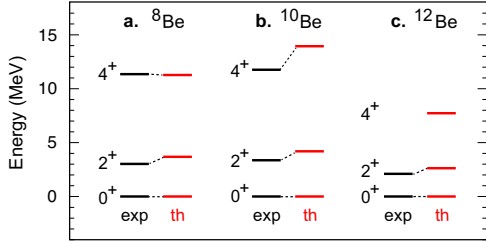

**Fig. 2 Excitation level energies of Be isotopes.** Theoretical levels ("th") are shown for **a** $^8$Be, **b** $^{10}$Be, and **c** $^{12}$Be, in comparison to experiment ("exp") taken from the National Nuclear Data Center's "Evaluated Nuclear Structure Data File" (http://www.nndc.bnl.gov/ensdf/).

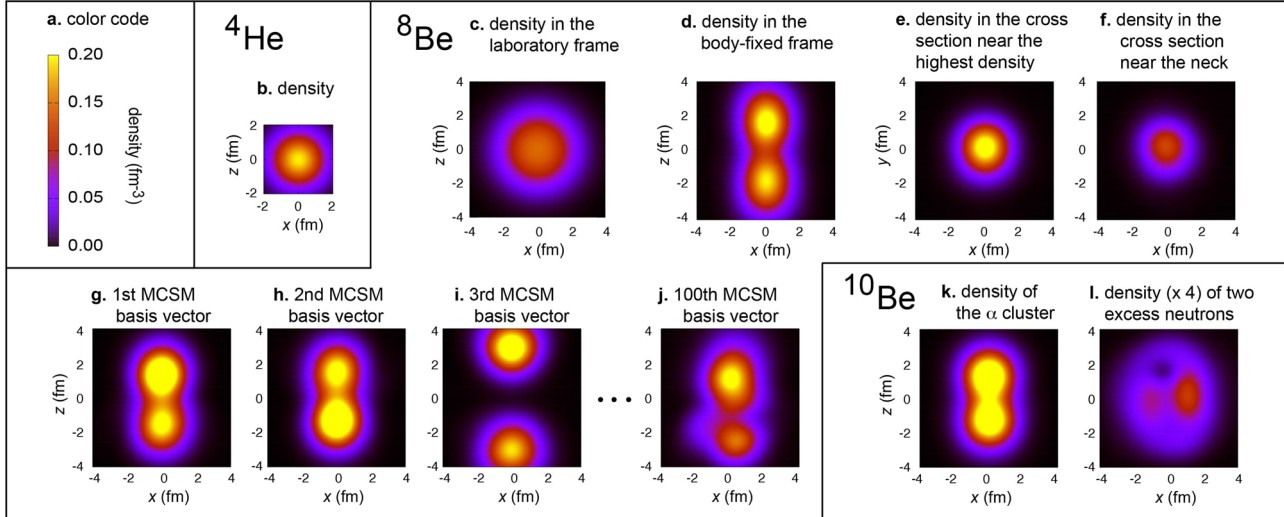

**Fig. 3 Density profiles of $^{8,10}$Be ground state in the body-fixed frame unless otherwise specified. a** Legend. **b** Matter density of $^4$He ($\alpha$ particle). **c** Matter density of $^8$Be in the laboratory frame. **d–f** Matter density $^8$Be on the $xz$ plane (d) and on the $xy$ plane (**e**, **f**). **g–j** Matter density of MCSM basis vectors for $^8$Be. **k** Matter density of the $\alpha$-cluster part of $^{10}$Be. **l** Density of the excess neutrons of $^{10}$Be.

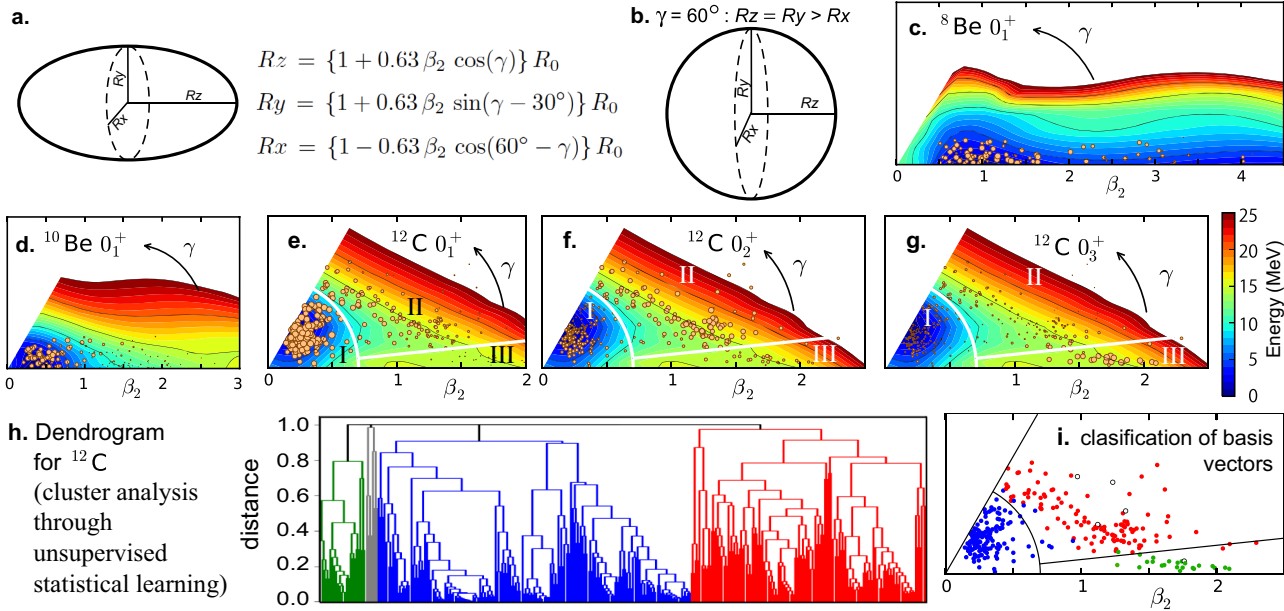

**Fig. 4 Classification of MCSM basis vectors by T-plot and Statistical Learning methods. a** Definition of parameters. The ellipsoid shows a prolate shape ($\gamma = 0$ or $R_y = R_x$). **b** An oblate shape. **c, d** Potential energy surface (PES) and T-plot for the ground state of $^{8,10}$Be. **e–g** PES and T-plot for the $0^+_{1,2,3}$ states of $^{12}$C with the decomposition to regions I–III. **h** MCSM basis vectors analyzed by the dendrogram of cluster analysis through unsupervised statistical learning. **i** T-plot circles classified by the same color as in **h**.

within bound state approximation (i.e., no explicit inclusion of continuum states), the $\alpha$ clustering appears just above the $\alpha$ decay threshold[5]. The laboratory frame density distribution (see Fig. 3c) is, in contrast, isotropic because of $J^\pi = 0^+$, blurring the clustering.

The $\alpha$ clustering of $^{10}$Be arises similarly. Figure 3k depicts twice the proton density for the ground-state Q-aligned state. This is expected to represent the matter density due to $\alpha$ clusters, as protons and neutrons tightly bind each other in the $\alpha$ cluster. The two $\alpha$ clusters are closer than in $^8$Be, because the additional two neutrons, called excess neutrons, behave like electrons making the covalent bond of a molecule. Figure 3l exhibits the density distribution of the two excess neutrons on the $xy$ plane at $z = 0$, indicating their circular motion about the $z$ axis. Note that such a molecular structure arises from first principles.

As shown in Fig. 4a, the lengths of $R_z$, $R_y$ and $R_x$ are parametrized[44] by variables, $R_0$, $\beta_2$, and $\gamma$, where $R_0$ is the average of $R_{x,y,z}$, and $\beta_2$, called *deformation parameter*, represents the magnitude of the ellipsoidal deformation from the sphere ($\beta_2 = 0$). The angle $\gamma$ ($= 0° - 60°$) specifies the ratio between $R_y$ and $R_x$: $\gamma = 0°$ (60°) implies prolate (oblate) shape (see Fig. 4a, b).

We use ($\beta_2, \gamma$) as partial but useful labeling of the basis vector. The $\beta_2$-$\gamma$ plane is introduced as usual (see Fig. 4c as an example): for a given pair ($\beta_2, \gamma$), the corresponding point on the plane is located at the distance $\beta_2$ from the origin, and at the angle $\gamma$ from the horizontal axis ($\gamma = 0$). In the T-plot of the MCSM[45], basis vectors are each shown by a circle, called T-plot circle, located at their respective ($\beta_2, \gamma$) values on the plane. The so-called potential energy surface (PES) is superposed on this plane: the PES represents the Hartree-Fock energies where the shapes are constrained to the ($\beta_2$, $\gamma$) values. The PES depicts what parts of the $\beta_2$-$\gamma$ plane gain more binding energy within the mean-field estimate. The importance of each basis vector can be evaluated by the overlap probability with the eigenstate currently considered, and is expressed by the area of each T-plot circle, meaning that larger circles are more relevant to the eigenstate. The T-plot is shown in Fig. 4c, d for the ground states of $^{8,10}$Be isotopes. In Fig. 4c, large T-plot circles are found around $\beta_2 = 1$, whereas smaller circles are scattered. Since $\beta_2 = 1$ and $\gamma = 0$

imply $R_z/R_{x,y} \sim 2.4$, this concentration of large T-plot circles suggests a strong stretching consistent with the dumbbell-like $\alpha$ clustering (Fig. 1b). The far-reaching bottom of the PES (dark blue area) is indicative of the slightly unbound nature of $^8$Be, while T-plot circles there are smaller. The T-plot circles move to smaller $\beta_2$ values in $^{10}$Be, again consistent with the weakening of $\alpha$ clustering and the shift to more spherical nuclear shapes due to tighter binding.

**Clustering in carbon-12 nucleus.** We here discuss the structure of $^{12}$C with the Daejeon16 interaction[37]. Figure 5a shows calculated lowest energy levels in comparison with experiment as well as comparisons of electric quadrupole (E2) and monopole (E0) transition strengths.

Figure 5b shows the convergence patterns of the energy eigenvalues. The MCSM calculation becomes more accurate by increasing the number (denoted $k$) of basis vectors. This means that the ground-state energy for a given $k$, denoted $E_k$, is lowered as $k$ increases. This work takes $k$ up to 300, partly because it is almost the limit set by the computer but mainly because a reasonable convergence is achieved. As $E_k$ converges slowly as a function of $k$, we monitor the convergence in a different way. We introduce the energy variance $r_k$, which is a measure of the difference between the exact eigenvalue and the approximate eigenvalue at $k$ (for details see Methods)[47]. This $r_k$ is a useful calculable variable: it is positive definite, and vanishes when the approximation becomes exact. We thus plot $E_k$ against $r_k$ in Fig. 5b. This figure also shows an extrapolation with a polynomial of $r_k$ up to a quadratic term. The estimated values at zero-variance are shown in Fig. 5a. The energy eigenvalues of the $0^+_1$ and $2^+_1$ states follow parallel trajectories for $r_k < 400$ MeV$^2$, suggesting that the excitation energy, $E_x(2^+_1)$, is estimated more accurately than individual eigenvalues.

The squared transition strength, $B(E2; 2^+_1 \to 0^+_1)$, and the $2^+_1$ spectroscopic electric quadrupole moment are calculated from the wave functions thus obtained, and point to an oblate shape with $\beta_2 \sim 0.6$. Electric charges are simply bare values: $1e$ for proton and $0e$ for neutron with $e$ being the unit charge. It is remarkable that

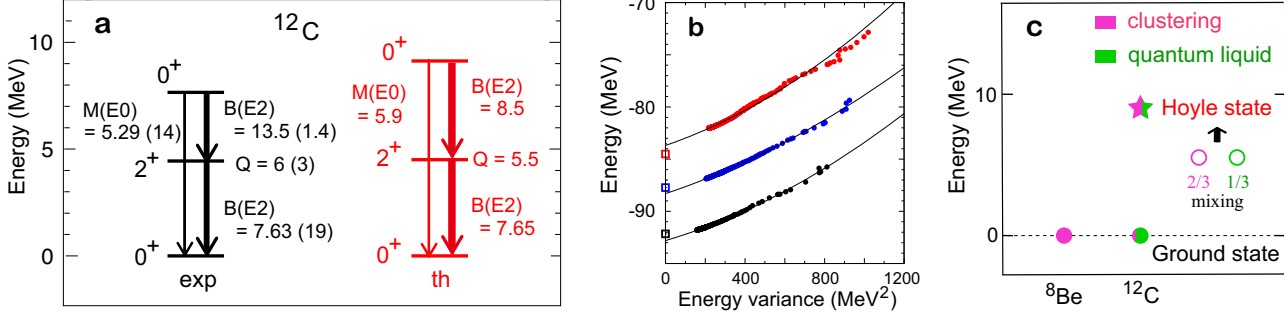

**Fig. 5 Properties of $^{12}$C nucleus. a** The $2_1^+$ and $0_2^+$ (Hoyle state) energy levels, and the $B(E2)$ ($M(E0)$) values in the unit of $e^2$ fm$^4$ (e fm$^2$) compared to experiments[20,46]. Data are also from the National Nuclear Data Center's 'Evaluated Nuclear Structure Data File' (http://www.nndc.bnl.gov/ensdf/). **b** Energy eigenvalues of the $0_1^+$ (black), $2_1^+$ (blue), and $0_2^+$ (red) states against the energy variance. Open squares indicate experimental values. Solid lines imply polynomial extrapolation. **c** Schematic illustration for the Hoyle state (star), comprising clustering (pink), and quantum–liquid (green) components (open circles). Ground-state properties are depicted for $^8$Be and $^{12}$C.

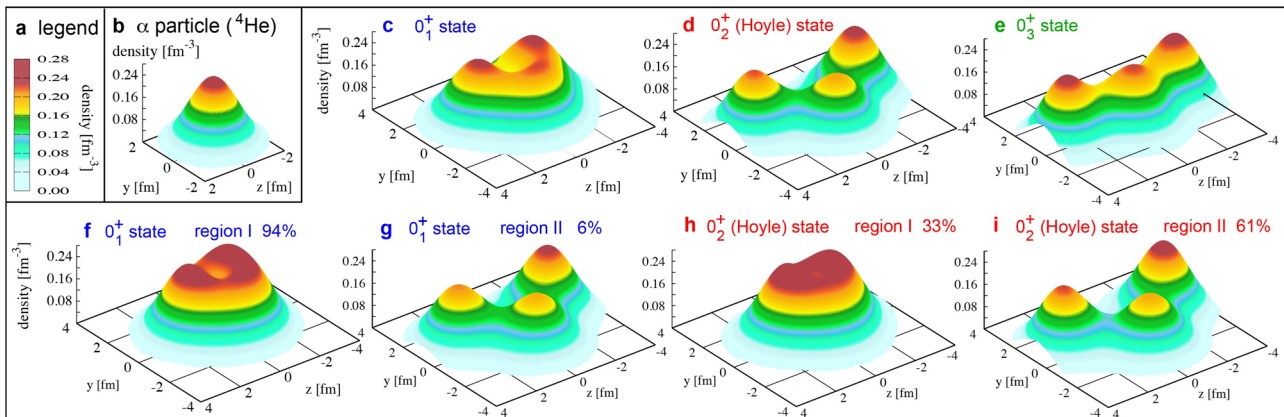

**Fig. 6 Density profiles on the $yz$ plane of $\alpha$ or $^{12}$C nuclei. a** Color code of the density. **b** Density of the $\alpha$-particle ground state. **c–e** Density of $0^+$ states of $^{12}$C nucleus. **f–i** Decomposition into the regions. The probability in the indicated region is shown.

$E_x(2_1^+)$, $B(E2; 2_1^+ \to 0_1^+)$ and the quadrupole moment are in excellent agreement with the experiment. We stress that the states of strong ellipsoidal oblate deformation, with $\beta_2 \sim 0.6$, can now be described in such an ab initio approach, with virtually all relevant correlations explicitly treated (i.e. no in-medium corrections like effective charges, effective operators, etc).

Following the cases of Be isotopes, we analyze the density distribution of the $0_1^+$ state in terms of Q-aligned states. Figure 6c shows its density profile on the $yz$ plane. For comparison, Fig. 6b displays the calculated density of the $\alpha$ particle. The peak values are similar between panels b and c. The pattern of Fig. 6c resembles the one in Fig. 1c–i. Three ($\alpha$-like) clusters are close-lying in both panels. In the former, the distances between the nearest peaks are ~1.9 and ~2.4 fm (if preferred, see two-dimensional plot in Supplementary Figure 1). These are smaller than the distance for $^8$Be (~3.5 fm), and this structure looks like Fig. 1c–i, being closer to a quantum liquid (i.e., normal nuclear matter with a basically constant nucleon density) rather than well-separated $\alpha$ clusters. The lower density region in the center of the nucleus (Fig. 6c) is seen. Although this contradicts the naive independent particle model with the filling of the lowest $s_{1/2}$ orbit, this trend is consistent with experiment[43].

**Novel picture of the Hoyle sate**. The $0_2^+$ state of $^{12}$C is called the *Hoyle* state[21]. Figure 6d shows its snapshot density profile obtained from the corresponding Q-aligned state, presenting clear differences from panel c. The Hoyle state appears to comprise three well-separated $\alpha$(-like) clusters. However, this is not the full story.

The features of the Hoyle state can be clarified by the T-plot shown in Fig. 4e, f for the $0_{1,2}^+$ states. These T-plot circles are widely distributed, in contrast to Be cases (Fig. 4c, d). In order to look into such a spread in the T-plot, we divide the whole PES plane into three regions, I, II and III. The region I is bound by $\beta_2 < 0.7$, as shown by arcs in Fig. 4e–g. The outer area is divided into region II for $6° \le \gamma \le 60°$ and region III for $0° \le \gamma \le 6°$, as separated by the outgoing straight lines in Fig. 4e–g.

Regarding the $0_1^+$ state, large T-plot circles in the region I seem to dominate the character of the $0_1^+$ state. In order to quantify this feature, we decompose the $0_1^+$ state into the region I, -II, and -III components comprising, respectively, basis vectors in the regions I, II, and III. Proper orthogonalization is performed among them (for technical details, see Methods). It is shown that the $0_1^+$ state lies in region I (II) with 94% (6%) probability, meaning that this state is predominantly in region I. Figure 6f exhibits the snapshot density profile obtained from the region I component of the Q-aligned state. The peak area of this density profile is flat and wide, like normal nuclear matter, which is a quantum liquid. This density profile shows an oblate and somewhat triangular shape similar to Fig. 1c (this may be seen better in the two-dimensional plot in Supplementary Figure 1). The density of the flat part is close to the central density of the $\alpha$ particle, higher than the normal density (~0.16 fm$^{-3}$). The implication of this common feature is worth noticing, as a possible characteristic feature of light nuclei.

The region II contribution to the snapshot density profile is exhibited in Fig. 6g, displaying three $\alpha$-like clusters. The region II thus implies the clustering, making a minor part (6%) of the $0_1^+$ state.

The Hoyle state shows different characters: it comprises, respectively, the region I, II, and III components with probabilities 33%, 61%, and 6%. Figure 6h, i separately displays the snapshot density profiles from regions I and II, where region III is omitted because of its minor contribution. The density profiles for regions I and II depict, respectively, the (quantum) liquid and the clustering patterns. Major basis vectors in region II are of triangular configurations, showing the special importance of the triangles (see Supplementary Figure 2 and Methods for more details). The Hoyle state thus comprises three $\alpha$-like clusters (Fig. 6i) with the probability ~2/3, but comprises, with the probability 1/3, the (quantum) liquid in a modestly ellipsoidal shape of $\beta_2 \sim 0.3$ (Fig. 6h). This is a very striking feature, because the clustering of the Hoyle state has been considered to emerge as a (almost) pure stand-alone mode[1–15]. Note that three peaks in panels d and i resemble that of panel b, with deeper valleys in panel i than those in panel d.

Figure 5c schematically shows how the Hoyle state is formed. While the ground state of $^8$Be is made of clusters, the ground state of $^{12}$C is mainly a liquid state with a certain mixing of clustering state, as shown by the pink-green circle. We mention that 6% probability implies amplitude ~1/4, which is not negligible in some cases. In fact, the structure of the Hoyle state is determined not only by the $NN$ interaction but also by the orthogonality to the $0_1^+$ state with this mixing. In Fig. 5c, combining all relevant effects, the $0_2^+$ state (star symbol) emerges as composed of clustering (region II) and quantum-liquid (region I) components (open circles in designated colors) with the probabilities ~2/3 and 1/3, respectively.

The quantum liquid and clustering were sometimes regarded as two "phases", and a phase transition was discussed[48,49]. The present work indicates that the nuclear forces mix the two "phases" attractively (repulsively) in the ground (Hoyle) state, which is incompatible with the phase transition picture. Instead, the crossover is a more appropriate concept with varying mixing for different states (see Fig. 5c). This is not like the usual crossover picture, partly because the orthogonality matters in excited states. Thus, ground and Hoyle states of $^{12}$C provide an unexplored facet to the physics of crossover[27].

The present ab initio no-core $NN$ interactions contain more intermediate and short-range components in general than other interactions for nuclear structure studies, and hence are more suitable for describing the coupling between states of two different characters; clustering and liquid. In contrast, an effective interaction designed just for the liquid is likely too soft to describe this coupling. The mixing of $\alpha$-clustering into the ground state is probably related to the $\alpha$ decay of heavier nuclei, the mechanism of which remains an open problem.

We note that ab initio nuclear forces favor triangular ($\alpha$-) configurations. Figure 6 (also Supplementary Figure 1) shows that in the Hoyle state, the largest angle of the triangle is slightly larger than 90°, whereas the ground state is in a nearly equilateral triangle shape with collapsed clustering.

The radius of the $0_1^+$ state is discussed in detail[35] in agreement with experiment[37,43], whereas it is overestimated in some other works[12]. The calculated density distribution of the ground state in the laboratory frame consistently depicts basic similarities to the experimental one[43] and the one calculated by the GFMC[18].

The difference of root-mean-square radii between the ground and Hoyle states becomes 0.36 fm, which is rather close to the value from experiment[50], ≈0.5 fm, compared with other theoretical values[50], 1.1–1.9 fm. The present smaller radius may signal some impact on reaction rates, e.g., of stellar triple-$\alpha$ fusion[24,25].

A linear-chain state[3,15] (see Fig. 1c) presently appears as the $0_3^+$ state with excitation energy ~14 MeV. Figure 6e shows its density profile, and Fig. 4g shows the T-plot. This excitation energy should come down in future calculations by expanding the model space. We simply stress the natural appearance of the linear-chain state.

**Implications and future directions**. The $\alpha$ clustering is discussed in terms of quantum many-body simulations from first principles, by using state-of-the-art supercomputing facilities[39,40]. The clustering in $^{8,10}$Be is clarified. Regarding $^{12}$C, its ground and first $2^+$ states appear as the members of a practically perfect rotational band with strong oblate deformation. This means that an ab initio interaction capable of describing enhanced quadrupole collectivity in nuclei is already available[37,38]; an absolutely encouraging message. These states are basically of a quantum liquid, with certain mixture of the clustering.

The Hoyle state, critical to nucleosynthesis and the origin of carbon-based life, exhibits the density profile with three $\alpha$ clusters. This is of great importance and portends its formation in the triple-$\alpha$ fusion process. Significantly, our analysis suggests that this state comprises quantum-liquid and clustering components in probability ratio ~1:2.

The structure of $^{12}$C can be viewed as a crossover of the quantum liquid and clustering, both of which are favored by ab initio nuclear forces, with weaker binding gained through the clustering. Inspired by this feature, the clustering component is expected to be meaningfully contained in a wide variety of nuclei and states with various forms and degrees, which may result in noticeable $\alpha$-emission, $\alpha$-knockout, or $\alpha$-decay, if appropriate. The clustering is presently considered to occur basically due to nuclear forces, without being a near-$\alpha$-threshold effect[5]. If this holds, the clustering can be a major component of well-bound excited states in some cases, for instance, where quantum-liquid states of a given spin/parity are lying higher in energy. This is an intriguing future ab initio challenge.

The present picture of $^{12}$C is obtained from the T-plot of MCSM basis vectors, and furthermore, it is verified by the cluster analysis (dendrogram) from statistical learning (see Discussion). This approach reduces the model-dependence in the crossover argument, and may open new avenues for looking into the physical content of quantum many-body wave functions emerging from complex CI calculations.

We finally note that the MCSM brought us crucial basis vectors for the Hoyle state in the plateau of the PES (i.e., region II), illuminating MCSM's superb capabilities. In this respect, the present work makes a prominent landmark in the MCSM achievement[28–31].

## Discussion

The present analysis based on the T-plot is examined from a completely different and more mathematical method: cluster analysis through unsupervised statistical learning[26]. The *objects* are basis vectors, and the *distance* between a pair of them, $\phi_i$ and $\phi_j$, is defined as $D(i,j) = 1 - |(\phi_i, \phi_j)|^2$, where the parenthesis means a scalar product (i.e., overlap integral) of two basis vectors with the $J^\pi = 0^+$ projection. Using this distance, a natural choice, we draw the dendrogram in the complete linkage framework shown in Fig. 4h: the dendrogram starts from the pair with the shortest distance near the bottom of Fig. 4h, and moves up linking more basis vectors having longer limits of the mutual distances (see Methods for more details). The threshold of the distance to define a *group* is set to 0.99, which means $|(\phi_i, \phi_j)| > 0.1$ within a given group. Figure 4h exhibits such classification

into four groups (shown by different colors). One of them is very minor and ignored. The remaining three groups in the dendrogram correspond remarkably well to regions I–III, as Fig. 4i displays them like T-plot in the same color code as Fig. 4h. The separation between regions I and II is particularly important in the present study, and the cluster analysis of the statistical learning clearly demonstrates that basis vectors in the region I are distinctly different from those in region II. It is striking that the classification conceived by the human brain (considerations on shapes or T-plot) can be supported by the statistical technique with no reference to shapes. This technique unravels the decomposition of 300 MCSM basis vectors referring only to "similarities" among them in the laboratory frame, by handling 44850 "data".

## Methods

**Monte Carlo Shell Model.** The MCSM[28–31] uses Slater determinants as the basis vectors, similarly to the conventional SM calculation (see Supplementary Note 3). However, the Slater determinants are not the same as those used in the conventional SM calculation where the Slater determinant is a direct product of some single-particle states in general. Note that a product of $A = Z + N$ single-particle states is taken for the no-core MCSM. In the case of the MCSM basis vector, each of such single-particle states is a superposition of all (more naive) single-particle states of the given model space, with amplitudes determined by stochastic and variational methods. The determination of those amplitudes is the most crucial part of the MCSM procedure. A basis vector for the MCSM calculation is a Slater determinant composed of such "stochastically—variationally deformed" single-particle states. Those single-particle states are mutually independent. By having a set of these MCSM basis vectors thus constructed, we diagonalize the Hamiltonian, and obtain energy eigenvalues and eigenfunctions.

The set of MCSM basis vectors are obtained one by one using the condition that the energy eigenvalue of the state of interest is lowered by the required amount or more, when adding each new basis vector. For each MCSM basis vector, the single-particle amplitudes mentioned above have to be determined properly. They are initialized by a stochastic process and are improved variationally. Because of the superposition over all single-particle states, the symmetries of the CI Hamiltonian are lost, and the projection onto the angular momentum, the parity, etc. is carried out for the MCSM basis vector. This is performed in the selection process of each basis vector. Many candidates for this basis vector are tried and rejected, keeping only ones meeting the minimum contribution condition. There are a number of practical methodological refinements omitted for brevity here. Many applications retain 50–100 MCSM basis vectors. The spurious center-of-mass motion is suppressed by the Lawson method[51]. The HO quanta of the center-of-mass motion are always monitored, and have been confirmed to be sufficiently small.

A large number of MCSM calculations have been performed as exemplified in refs. [32,33,45,52–58]. Most of these applications feature the adoption of an inert core with valence nucleons for the description of the nuclear properties. However, here we implement the ab initio no-core MCSM. All nucleons are then activated, and the number of single-particle orbits is much larger. Furthermore, the adopted realistic NN interactions fitting two-nucleon scattering data feature strong short-range and tensor components coupling single-particle states more substantially. Thus, solving the ab initio no-core MCSM is far more computationally challenging. We select 100 MCSM basis vectors for Be isotopes, and 300 for $^{12}$C nucleus. The maximum dimension of the vector space for a comparable full CI calculation is $1.2 \times 10^{12}$ for $^{8}$Be and $1.9 \times 10^{19}$ for $^{12}$C. The single-particle wave functions used in the present MCSM calculations are taken from the eigenstates of the HO potential, with $\hbar\omega$ being 15 and 20 MeV, respectively, for Be and C isotopes[35]. Because of the ab initio no-core feature, the results are not sensitive to the $\hbar\omega$ value within reasonable ranges[35].

**Convergence pattern of MCSM results.** The present CI calculation is performed by diagonalizing the Hamiltonian, denoted $H$, with a certain number of MCSM basis vectors. By increasing this number, called $k$, the lowest eigenvalue for a given quantum number is lowered.

In order to see the convergence of the calculated eigenvalue as $k$ increases, we use the variance[47]: $r_k = \langle\phi_k|H^2|\phi_k\rangle - \langle\phi_k|H|\phi_k\rangle^2$, where $\phi_k$ implies the eigenstate obtained by including from the first up to the $k$-th basis vectors. If $\phi_k$ represents the exact solution, $r_k = 0$ holds. We plot the eigenvalues against the variance $r_k$ instead of $k$ (see Fig. 5b). Indeed, as $k$ increases, the energy eigenvalue for a given quantum number is lowered, and $r_k$ basically decreases, approaching zero for sufficiently large $k$. As the MCSM solution becomes closer to the exact one, this behavior can be empirically simulated by some extrapolation method, such as polynomials in $r_k$. A quadratic polynomial is used in this work.

**Q-aligned state and density profiles.** The eigenstate, with spin/parity $J^\pi$ and other quantum numbers $\xi$, calculated by the MCSM is expressed as a superposition of MCSM basis vectors mentioned above as

$$\Psi(J^\pi, \xi) = \mathcal{N} \sum_i f_i^{(0)}(J^\pi, \xi) \hat{P}(J^\pi) \phi_i^{(0)}, \quad (1)$$

where $\mathcal{N}$ denotes a normalization constant, $i$ is the index of the basis vector, $\phi_i^{(0)}$ and $f_i^{(0)}$ mean, respectively, the $i$-th basis vector and its amplitude. Here, $\hat{P}(J^\pi)$ implies a projector onto the quantum numbers such as $J^\pi$. The additional quantum number $\xi$, for instance the sequential index, can be omitted hereafter unless needed.

If the orientation of $\phi_i^{(0)}$ in the three-dimensional configuration space is changed (i.e. rotated), the same eigenstate is obtained with appropriately changed $f_i^{(0)}$, because of the projection by $\hat{P}$. However, for the snapshot state of a given rotational band, the orientations of individual basis vectors matter, as discussed in Results when referring to Fig. 3g–j.

As a general trend, in order to gain in binding energy, the attractive feature of the NN interaction is expected to maximize the overlap between ellipsoids corresponding to individual basis vectors. This implies alignment of the longest axis ($R_z$) of each MCSM basis vector to the same direction. The $R_y$ and $R_x$ axes are aligned likewise. In Fig. 3g–j, the density profiles are shown for selected MCSM basis vectors thus aligned.

We then introduce the Q-aligned state: all basis vectors are aligned in this way, and are superposed with appropriate amplitudes so that its projection onto $J^\pi = 0^+$ becomes the MCSM ground state. From the aforementioned general argument, this Q-aligned state is expected to show the features of the snapshot (intrinsic) state to a good extent. Furthermore, a more detailed inspection presented later indicates that the picture of a rotating object, classical in some sense, holds nearly perfectly, which is a non-trivial and even unexpected feature for an ab initio calculation. The "snapshot" we seek from theory can thus be provided by the density profile of the Q-aligned state.

We then rotate $\phi_i^{(0)}$ with appropriate Euler angles so that the ellipsoidal axes of the resulting basis vector $\phi_i$ are aligned to the pre-fixed directions (as mentioned in Results). This is done for all $i$'s separately. The Q-aligned state is then defined by

$$\Omega(J^\pi, \xi) = \mathcal{N}' \sum_i f_i(J^\pi, \xi) \phi_i, \quad (2)$$

where $\mathcal{N}'$ denotes a normalization constant and $f_i$ stands for modified amplitude. As the state $\Omega(J^\pi, \xi)$ is defined in the body-fixed frame, it does not conserve $J^\pi$, but the amplitude $f_i(J^\pi, \xi)$ retains dependence on $J^\pi$ and $\xi$. The same eigenstate as in eq. (1) is written as

$$\Psi(J^\pi, \xi) = \mathcal{N} \sum_i f_i(J^\pi, \xi) \hat{P}(J^\pi) \phi_i. \quad (3)$$

Figure 3g–i depict the density profiles of basis vectors thus aligned, and Fig. 3d–f exhibit the density profiles of the Q-aligned state (see eq. (2)) for the $^{8}$Be ground state. Figure 3k–l present the density profiles for the $^{10}$Be ground state.

Likewise, Figures 6c–e exhibit the density profiles of the Q-aligned states for the three $0^+$ states of $^{12}$C.

We discuss the validity of the Q-aligned state as the snapshot state, from which the snapshot of the nucleus can be obtained. The Q-aligned state in eq. (2) is obtained from a given $J^\pi = 0^+$ MCSM eigenstate, like the state in eq. (1). We generate a $J^\pi = 2^+$ state from this Q-aligned state. Here, we introduce the quantum numbers $M$ and $K$. The former means, as usual, the z-component of the angular momentum $\vec{J}$ in the laboratory frame. The $K$ quantum number is defined in the body-fixed frame, implying the z- (x-)component of $\vec{J}$ for prolate (oblate) shape. In the case of $J = 0$, only $M = K = 0$ is allowed. For $J = 2$, $M$ can be any value between $-2$ and 2, on which the energy does not depend due to the rotational symmetry of the Hamiltonian. In the limit of an ideal quantum rotor, the $J^\pi = 0^+$ and $2^+$ members are obtained by rotating the same snapshot (or intrinsic) state (i.e., by the angular momentum projection of this state). The Q-aligned state, by definition, generates the $J^\pi = 0^+$ MCSM eigenstate. This means that if the $J^\pi = 2^+$ state is also generated from the Q-aligned state in the same way as the one for the $J^\pi = 0^+$ state, the rotational band picture with a good $K$ holds. Here, the same way implies the projection onto $J^\pi = 2^+$ with $K^\pi = 0^+$. This hypothesis is examined by calculating the overlap probability between the $J^\pi = 2^+$ state thus obtained and the corresponding $J^\pi = 2^+$ MCSM eigenstate. The resulting overlap probabilities are 99% for both $^{8}$Be and $^{12}$C. These values definitely suggest that for the lowest $J^\pi = 0^+$ and $2^+$ states of these nuclei, the corresponding Q-aligned state is regarded as the snapshot (or intrinsic) state, and the resulting nuclear snapshot makes nearly perfect sense. The overlap probability for $^{10}$Be turns out to be 90%, which, though large, is smaller than the one for $^{8}$Be or $^{12}$C. The difference from $^{8}$Be or $^{12}$C is due to excess neutrons, and means that this snapshot is most important but, to be complete, other snapshots may arise with 10% probability in total. The effects of excess neutrons are an interesting future subject.

The analysis presented above is not currently feasible for the Hoyle state of $^{12}$C due to the computing limitations and we present a different argument later.

We next discuss the density profiles of the decomposed Q-aligned states. For the sake of simplicity, we restrict ourselves to the $J^\pi = 0^+$ states. The decomposed Q-aligned state corresponding to the region L (=I, II, or III) of the PES is defined

as,

$$\theta_n^{[L]} = \mathcal{N}'_{L,n} \sum_{i \in RegionL} f_i(0_n^+) \phi_i, \tag{4}$$

where $\mathcal{N}'_{L,n}$ is a normalization and $n$ refers to the $n$-th $J^\pi = 0^+$ state. The projection onto $J^\pi = 0^+$ produces,

$$\Theta_n^{[L]} = \mathcal{N}''_{L,n} \hat{P}(J^\pi = 0^+) \theta_n^{[L]}, \tag{5}$$

where $\mathcal{N}''_{L,n}$ is a normalization.

For the $J^\pi = 0_1^+$ state ($n = 1$), this definition works for region I. The density profile of $\theta_{n=1}^{[I]}$ is shown in Fig. 6f, exhibiting a feature close to quantum liquid. We calculate the overlap probability between $\Theta_1^{[I]}$ and the $J^\pi = 0_1^+$ MCSM eigenstate. The obtained value, 94%, is mentioned in Results, as an indicator of region I dominance of the $J^\pi = 0_1^+$ state, consistent with the visual appearance in Fig. 6f.

For region II, however, the situation is slightly more complex: states $\theta_1^{[I]}$ and $\theta_1^{[II]}$ have a small overlap, and hence $\Theta_1^{[I]}$ and $\Theta_1^{[II]}$ are not completely orthogonal. We take a superposition of $\Theta_1^{[I]}$ and $\Theta_1^{[II]}$, so that the resulting projected state, denoted $\tilde{\Theta}_1^{[II]}$, is orthogonal to $\Theta_1^{[I]}$, fixed above. By using the same relative mixing amplitudes, $\tilde{\theta}_1^{[II]}$ is obtained and normalized, and its density profile is shown in Fig. 6g, indicating $\alpha$-like clustering. The overlap probability of $\tilde{\Theta}_1^{[II]}$ with the $J^\pi = 0_1^+$ MCSM eigenstate is 6%, suggesting a minor mixture of the clustering into the $J^\pi = 0_1^+$ state. We analyzed region III similarly, but found the overlap probability to be negligible.

For the $J^\pi = 0_2^+$ (Hoyle) state, we first take the region II component (i.e., not orthogonalized to any state yet), because of its importance: the $\Theta_2^{[II]}$ state shows the overlap probability with $J^\pi = 0_2^+$ (Hoyle) state, 61%, meaning that the region II is most relevant to the $J^\pi = 0_2^+$ (Hoyle) state. The density profile of $\theta_2^{[II]}$ is shown in Fig. 6i, depicting a clear $\alpha$-like cluster structure. Similar to the previous case, a superposition of $\theta_2^{[II]}$ and $\theta_2^{[III]}$ and another superposition of $\Theta_2^{[II]}$ and $\Theta_2^{[III]}$ are taken, so that the resulting state, denoted $\tilde{\Theta}_2^{[III]}$, is orthogonal to $\Theta_2^{[II]}$. This order of the orthogonalization was taken because of a relatively large overlap probability of $\Theta_2^{[III]}$ with the $J^\pi = 0_2^+$ state, but this is due to the non-orthogonality between $\Theta_2^{[III]}$ and $\Theta_2^{[II]}$. In fact, the overlap probability of $\tilde{\Theta}_2^{[III]}$ with the $J^\pi = 0_2^+$ state becomes as small as 6%. We next obtain $\tilde{\theta}_2^{[I]}$ and $\tilde{\Theta}_2^{[I]}$, so that $\tilde{\Theta}_2^{[I]}$ is orthogonal to $\Theta_2^{[II]}$ and $\tilde{\Theta}_2^{[III]}$. The overlap probability of $\tilde{\Theta}_2^{[I]}$ with the $J^\pi = 0_2^+$ (Hoyle) state is 33%, which is remarkably high. Figure 6h shows the density profile of the $\tilde{\theta}_2^{[I]}$ state thus obtained, and implies that this component of the Hoyle state is basically a quantum liquid. We now see that the cluster and the quantum liquid are strongly coupled and mixed in the Hoyle state, as stated in the Results. The density profile obtained from $\tilde{\theta}_2^{[III]}$ is shown in Supplementary Figure 1j, where a linear configuration like Fig. 1c–iii is seen.

**Density profiles of basis vectors and Hoyle state**. Supplementary Figure 2 shows the density profiles for selected MCSM basis vectors in region II. Two basis vectors, carrying the two largest overlap probabilities with the Hoyle state, are shown for each 6°-bin from $\gamma = 6°$ to 60°. Here, the overlap is calculated with the $J^\pi = 0^+$ projection and normalization. The left basis vector in panel e carries the largest probability, 56%, among all shown in Supplementary Figure 2. This basis vector depicts 85% overlap probability with the (normalized) region II component of the Hoyle state. The region II Q-aligned state is thus dominated by a single Slater determinant. Consistently, Supplementary Figure 1i is similar to Supplementary Figure 2e (left). Panels c–f display distinct $\alpha$ clustering, but the clustering is smeared in the Q-aligned state due to the remaining basis vectors (panels g–k). This is due to the fluctuations of the cluster configurations, and such fluctuations are related to the gain of correlation energies provided by the $NN$ interaction. The fluctuations imply that the snapshot is not completely fixed, and somewhat varies around the one given by the Q-aligned state. The dominance of the basis vector in Supplementary Figure 2e (left) certainly overshadows this fluctuation.

We emphasize that apart from minor differences, all basis vectors in panels c–k show triangular clustering configurations in common, reflecting the underlying importance of triangular configurations in region II.

We note that the basis vector in Supplementary Figure 2e (left) is the 3rd basis vector in the sequence of the present MCSM basis-vector generation process, meaning that such an important basis vector is picked up at this very early stage and that the calculation then proceeded on a more gradual process of further lowering the energy.

The Q-aligned state for region I is closer to spherical shape and lacks visible clustering. It is interesting to note that this result underscores the capability of our methods to reveal significant non-collective aspects of our solutions in conjunction with the collective aspects.

The region I and -II components are mixed in the Hoyle state, as emphasized in the Results. The signs of the mixing amplitudes are determined also by the orthogonality to the ground state, and the $NN$ interaction between these components yields a repulsive contribution. Thus, this off-diagonal contribution cancels substantial $\alpha$-$\alpha$ correlation effects, which tend to lower the energies of

clustering states. If this aspect were overlooked, the $\alpha$-$\alpha$ interaction could look erroneously weak.

**Dendrogram of MCSM basis vectors**. The dendrogram of basis vectors shown in Fig. 4h is drawn in the complete linkage framework of the clustering from statistical learning[26]. The distance defined in the Discussion is now denoted $d(x, y)$, where $x$ and $y$ are elements, and are basis vectors with the $J^\pi = 0^+$ projection and normalization in the present work.

The quantity $L$ is introduced as the longest distance for a given set $\{a, b, c, ....\}$: $L(a, b, c, ....) = max\{d(a, b), d(a, c), d(b, c), ....\}$. A set $\{a, b, c, ....\}$ is defined by the threshold $t$ so that $L(a, b, c, ....) < t$ is fulfilled. In Fig. 4h, the dendrogram is drawn from the minimum $t$, for which the pair, $\{a, b\}$, can form the first set. By raising $t$, another basis vector $c$ can join and the second set, $\{a, b, c\}$, is formed, because of $d(a, b)$, $d(a, c)$, $d(b, c) < t$. We can continue by adding elements for larger $t$ values, and draw the dendrogram. In Fig. 4h, $t$ is changed from a small value up to 0.99. If the distance is beyond the threshold, no additional element is added, and the group is fixed. We end up with three major groups and a minor group, as shown in Fig. 4h. The minor one is not considered hereafter. The three major groups basically correspond to different regions determined by the T-plot shown in Fig. 4i. This clean relation is remarkable, as this dendrogram classification is carried out without knowing the shape of each basis vector.

In the MCSM, basis vectors are added so that the energy eigenvalue is lowered. The newly added basis vector can be very different from existing basis vectors or moderately different from them. The former and the latter may belong to different groups, and the present cluster analysis method unravels such grouping structure. In this sense, this method should have general applicability.

**Configuration convergence of the ground and Hoyle states of $^{12}$C**. The single-particle states used in the present MCSM calculation are taken from single-particle states of the HO potential, each of which has a fixed value for its HO quanta $(2n_r + l)$ with $n_r$ and $l$ being the number of radial nodes and the orbital angular momentum, respectively. By utilizing this property, the MCSM eigenfunction can be grouped into components with definite total quanta of the harmonic oscillator. The difference between a given total quanta and the lowest possible quanta is denoted, N$\hbar\omega$. So, the $0\hbar\omega$ component corresponds to the conventional SM states without excitations between HO shells (below or above). Supplementary Figure 3 depicts the probability of the N$\hbar\omega$ component as a function of N for several MCSM eigenstates of $^{12}$C nucleus. The ground and $2_1^+$ states have $0\hbar\omega$ components with probabilities ≈0.6. In contrast, the $0_2^+$ (Hoyle) state shows rather constant probabilities up to N = 8. This analysis clearly indicates characters drastically different between the ground and Hoyle states.

The gradually decreasing probabilities of the N = 2, 4, 6 components of the ground and $2_1^+$ states displayed in Supplementary Figure 3 suggest strong polarization effects, which are the origin of the effective charges needed in the conventional SM calculations with $0\hbar\omega$ or similar wave functions. The present result indicates that nucleons remain in the lowest HO shell with probabilities ~50 or 60% only, and the probabilities are damped gradually as N increases. However, this is not the case with the Hoyle state: the probability even increases towards N = 6 as a consequence of the $\alpha$ clustering. It is not damped quickly, implying that the inclusion of more single-particle orbits, corresponding to higher HO shells, may further improve the calculation. The convergence as a function of the number of HO shells included is an important technical issue. The number of HO shells is appropriate for the present purpose, and is also at the edge of the current computational feasibility. Relevant detailed discussions are found in ref. [35].

## Data availability
All data relevant to this study are shown in the paper, but if more details are needed, they are available from the corresponding author upon reasonable request.

## Code availability
Reasonable inquiries on the MCSM code are responded to by the corresponding author.

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

## Acknowledgements

The authors thank Dr. Toshio Suzuki for valuable suggestions on the SM studies on *p*-shell nuclei, and Drs. T. Kibédi, A. E. Stuchbery, and A. Gorgön for valuable discussions on experimental data of $^{12}$C. Useful discussions on the crossover with Drs. K. Fukushima and S. Uchino are appreciated. The MCSM calculations were performed on the super-computers K and Fugaku at RIKEN AICS (hp190160, hp200130, hp210165). This work was supported in part by MEXT as "Priority Issue on Post-K computer" (Elucidation of the Fundamental Laws and Evolution of the Universe) (hp160211, hp170230, hp180179, hp190160) and "Program for Promoting Researches on the Supercomputer Fugaku" (Simulation for basic science: from fundamental laws of particles to the creation of nuclei) and by JICFuS. This work was supported by JSPS KAKENHI Grant Numbers JP19H05145, JP21H00117, JP21K03564, JP20K03981, JP17K05433, and JP18H05462. J.P.V. and P.M. acknowledge support from US-DOE grants DE-FG02-87ER40371 and DESC-0018223.

## Author contributions

T.O. supervised the whole study and wrote the manuscript; T.A. performed many of the CI calculations; T.Y. performed some of the CI calculations and initiated the dendrogram analysis; Y.T. drew T-plot and performed crucial calculations; N.I. contributed to the discussions of clusters; Y.U., J.V., P.M., and H.U. contributed to various in-depth discussions; N.S. made the main part of the computer codes. All authors discussed the results and commented on the manuscript.

## Competing interests

The authors declare no competing interests.
