## [Peer Review File · Nature Communications]

α -Clustering in Atomic Nuclei from First Principles with Statistical Learning and the Hoyle State CharacterEditorial Note: This manuscript has been previously reviewed at another journal that is not operating a transparent peer review scheme. This document only contains reviewer comments and rebuttal letters for versions considered at *Nature Communications*.

REVIEWERS' COMMENTS

Reviewer #1 (Remarks to the Author):

I am very happy with the response of the authors' who have taken onboard my comments and criticism of the original manuscript. The authors have greatly improved the readability of the paper, which was my principle concern relating to publication.

The authors have undertaken considerable work to make many of the specialist technical sections more accessible to a general reader. Figure captions and text have been harmonised and details relevant to promote a fuller understanding have been included where appropriate. This includes details that were previously confined to the methods section.

I commend the authors for their careful and considered changes and the effort that has been put into creating a manuscript that, as I commented previously, will be an excellent resource with noteworthy results from this theoretical approach.

I am therefore happy for this manuscript to be published in Nature Communications.

Reviewer #2 (Remarks to the Author):

As I have stated in my first report for this paper, I believe that it presents a state of the art study of alpha-clustering in atomic nuclei. In the revised version submitted to Nature communications the authors have addressed my remarks from the first report and they have implemented appropriate modifications in the manuscript. Therefore, I would certainly recommend publication in Nature Communications.

Reviewer #4 (Remarks to the Author):

This paper on alpha-clustering in light nuclei is quite impressive and deserving of publication in Nature Physics. It uses realistic nucleon-nucleon interactions and a sophisticated configuration interaction calculation to provide a fine analysis of cluster structure, focusing on ^8Be and ^{12}C , including the famous Hoyle state in ^{12}C that is the gateway to carbon production in the universe. The detailed picture of ^{12}C states summarized in Fig.6 is particularly illuminating.

As the fourth reviewer, I gather the paper has been substantially altered from its original form, mostly in response to the comments of Referee #1, and is consequently undoubtedly improved. The paper is certainly of considerable interest to nuclear physics, and the methods for analysis of basis vectors of deformed states summarized in Fig.4 should be of interest to those doing CI calculations for atomic or molecular systems. Also, the characterization of ^{12}C states in clustering vs. quantum liquid terms is quite thought-provoking and of general interest.

I will add only a few comments which I leave as an option for the authors to consider, since the paper has already been under extended review.

1) The experimental excitation spectra for ^{10}Be and ^{12}Be shown in Fig.2 are considerably more complicated than illustrated. In particular, ^{10}Be has second $2+$ and $0+$ states around 6 MeV; although these are not of a rotational character, the second $0+$ may have some affinity with the Hoyle state in ^{12}C , in that they are both multi-particle sd -shell excitations that appear much lower in the spectrum than expected for p -shell states. Since ^{10}Be and ^{12}Be are not a focus of the paper, this is a relatively minor point, but the authors might note they have picked out only rotational states from more complex experimental spectra.

2) At the end of the Be clustering section where the T-plot for ^{10}Be , Fig.4d, is briefly discussed, the remark is made that T-plot circles moving to smaller β_2 values is consistent with the weakening of alpha clustering. This is true, but it's also simply the consequence of tighter binding of ^{10}Be vs. ^8Be and consequently a more spherical nucleus.

3) The authors note that the ^8Be cluster structure they obtain agrees well with earlier variational Monte Carlo calculations from Ref.23. However, they may have missed the Green's function Monte Carlo calculation for the ^{12}C ground and Hoyle states in Ref.24, which shows (Fig.20) one-dimensional density distributions that look consistent with the present results. Of course, the present paper gives a much more visceral picture of the clustering with the 3D images of Fig.6. The GFMC ground-state density matches that inferred from electron scattering experiments, so it would be interesting to know if the present ground state density also matches experiment (which is not easily seen from Fig.6) – this could just be stated in a few words.

Replies to reviewer's comments

We here show reviewer comments and our replies in the point-to-point format. Our replies and explanations are shown in purple with indentation, while the quoted revised text is shown in dark blue with double indentation. Black letters indicate reviewers' comments.

Reviewer comments:

Reviewer #4 (Remarks to the Author):

This paper on alpha-clustering in light nuclei is quite impressive and deserving of publication in Nature Physics. It uses realistic nucleon-nucleon interactions and a sophisticated configuration interaction calculation to provide a fine analysis of cluster structure, focusing on 8Be and 12C , including the famous Hoyle state in 12C that is the gateway to carbon production in the universe. The detailed picture of 12C states summarized in Fig.6 is particularly illuminating.

As the fourth reviewer, I gather the paper has been substantially altered from its original form, mostly in response to the comments of Referee #1, and is consequently undoubtedly improved. The paper is certainly of considerable interest to nuclear physics, and the methods for analysis of basis vectors of deformed states summarized in Fig.4 should be of interest to those doing CI calculations for atomic or molecular systems. Also, the characterization of 12C states in clustering vs. quantum liquid terms is quite thought-provoking and of general interest.

I will add only a few comments which I leave as an option for the authors to consider, since the paper has already been under extended review.

We thank Reviewer #4 for the very positive evaluation of our work. We have revised the paper following the suggestions, as shown below. The modified parts are shown in red in the attached pdf file of the revised paper.

1) The experimental excitation spectra for 10Be and 12Be shown in Fig.2 are considerably more complicated than illustrated. In particular, 10Be has second $2+$ and $0+$ states around 6 MeV; although these are not of a rotational character, the second $0+$ may have some affinity with the Hoyle state in 12C , in that they are both multi-particle sd-shell excitations that appear much lower in the spectrum than expected for p-shell states. Since 10Be and 12Be are not a focus of the paper, this is a relatively minor point, but the authors might note they have picked out only rotational states from more complex experimental spectra.

We revised manuscript by adding the following words after "... ground state" (lines 115-116)

... ground state, while other excited states are omitted for clarity.

2) At the end of the Be clustering section where the T-plot for ^{10}Be , Fig.4d, is briefly discussed, the remark is made that T-plot circles moving to smaller beta-2 values is consistent with the weakening of alpha clustering. This is true, but it's also simply the consequence of tighter binding of ^{10}Be vs. ^8Be and consequently a more spherical nucleus.

We revised manuscript by adding the following words after "... weakening of α -clustering" (lines 257-258)

...weakening of α -clustering and the shift to more spherical nuclear shapes due to tighter binding.

3) The authors note that the ^8Be cluster structure they obtain agrees well with earlier variational Monte Carlo calculations from Ref.23. However, they may have missed the Green's function Monte Carlo calculation for the ^{12}C ground and Hoyle states in Ref.24, which shows (Fig.20) one-dimensional density distributions that look consistent with the present results. Of course, the present paper gives a much more visceral picture of the clustering with the 3D images of Fig.6. The GFMC ground-state density matches that inferred from electron scattering experiments, so it would be interesting to know if the present ground state density also matches experiment (which is not easily seen from Fig.6) – this could just be stated in a few words.

We revised manuscript by adding the following sentences after "... other works¹²." (lines 444-447)

... other works¹². The calculated density distribution of the ground state in the laboratory frame consistently depicts basic similarities to the experimental one⁴³ and the one calculated by the GFMC¹⁸.

We hope that all the suggestions by Reviewer #4 have been incorporated into the revised manuscript, and that the present paper has become suitable for publication in *Nature Communications*.